# Immiscible hydrocarbon fluids in the deep carbon cycle

Fang Huang[1], Isabelle Daniel[2], Hervé Cardon[2], Gilles Montagnac[2] & Dimitri A. Sverjensky[1]

The cycling of carbon between Earth's surface and interior governs the long-term habitability of the planet. But how carbon migrates in the deep Earth is not well understood. In particular, the potential role of hydrocarbon fluids in the deep carbon cycle has long been controversial. Here we show that immiscible isobutane forms *in situ* from partial transformation of aqueous sodium acetate at 300 °C and 2.4–3.5 GPa and that over a broader range of pressures and temperatures theoretical predictions indicate that high pressure strongly opposes decomposition of isobutane, which may possibly coexist in equilibrium with silicate mineral assemblages. These results complement recent experimental evidence for immiscible methane-rich fluids at 600–700 °C and 1.5–2.5 GPa and the discovery of methane-rich fluid inclusions in metasomatized ophicarbonates at peak metamorphic conditions. Consequently, a variety of immiscible hydrocarbon fluids might facilitate carbon transfer in the deep carbon cycle.

[1] Department Earth & Planetary Sciences, Johns Hopkins University, 3400 N. Charles St., Baltimore, Maryland 21218, USA. [2] Université de Lyon, Université Lyon 1, Ens de Lyon, CNRS, UMR 5276 Lab. de Géologie de Lyon, Villeurbanne F-69622, France. Correspondence and requests for materials should be addressed to F.H. (email: fanghuang007@gmail.com).

Although most petroleum and natural gas in shallow reservoirs in Earth's crust is biogenic in origin, deep sources of abiogenic hydrocarbons have been suggested for some of these reservoirs[1–4]. Natural occurrences and experiments have shown that abiogenic methane and minor amounts of heavier hydrocarbons may have formed in the shallow crust through thermogenic breakdown of biomolecules, Fischer–Tropsch Type (FTT) synthesis from synthesis from CO or $CO_2$, and decarboxylation of aqueous acetate[5]. However, at the higher pressures and temperatures of the deep crust and upper mantle, a potential role for hydrocarbon fluids in the deep Earth carbon cycle has long been very controversial[1,2,4]. Methane has been found in fluid inclusions in metasomatized ophicarbonates[6] and diamonds[7]. Experiments have produced methane from carbonate[8,9] or from gas-water-rock reactions[10], ethane from methane[11] and heavy hydrocarbons from methane[12,13], but experimental evidence of the formation of abundant hydrocarbons heavier than methane in the presence of water and silicate rocks in the deep Earth is not known.

Recent experiments have demonstrated the formation of immiscible methane-rich fluids in the presence of an aqueous fluid at temperatures of 600–700 °C and 1.5–2.5 GPa (ref. 14), with trace amounts of heavier hydrocarbons detected. Traditional models of high pressure C-O-H fluids in the deep crust and upper mantle only consider species such as CO, $CO_2$, $CH_4$, $C_2H_6$, $H_2$, $O_2$ and $H_2O$ (ref. 15), not sufficient to understand the formation of heavier hydrocarbons. No specific process has been identified that can produce heavier hydrocarbons more abundant than methane in the presence of aqueous fluids. Nor have the relative importance of pressure and temperature on hydrocarbon fluid stability been clearly identified, although it has been suggested that the stability of heavy hydrocarbons should be favoured over methane at pressures above about 3.0 GPa (refs 4,8). Consequently, the potential role of deep abiogenic hydrocarbon fluids in the Earth's deep carbon cycle is still poorly understood and possibly underestimated.

Possible sources of carbon in abiogenic hydrocarbons at upper mantle pressures and temperatures include subducted organic matter and inorganic carbonates, and dissolved aqueous organic species. Acetate is likely to be one of the most important aqueous organic species to be subducted owing to its ubiquitous occurrence and apparent long-term persistence in oil-field brines in sedimentary basins[16], and its occurrence in pore waters in marine sediments[17,18] and in hydrothermal vents[19]. Acetate contains carbon with an overall oxidation state of zero. Consequently, acetate is also a convenient source of zero oxidation-state carbon in experimental studies.

Low-pressure experimental studies demonstrate that at 300 °C and 35 MPa, aqueous acetate will decarboxylate into $CO_2$ and $CH_4$, or will be oxidized by water to $CO_2$ and $H_2$ (ref. 20). However, recent theoretical studies using the Deep Earth Water model suggest that above about 3.0 GPa acetate could be thermodynamically stable with aqueous $CO_2$ and aqueous hydrocarbons under subduction-zone conditions[21]. Interestingly, a pressure threshold of ∼3.0 GPa was previously noted in experimental and theoretical studies of abiogenic hydrocarbon formation[4,8], but there are no previous high pressure experimental studies of aqueous acetate. Recent experiments starting with aqueous formate suggest that water and pressure might indeed stabilize hydrocarbon species (mainly methane) under deep Earth conditions[14].

Our experimental results at 300 °C and ∼3.0 GPa show that aqueous sodium acetate can be transformed into immiscible hydrocarbons, mainly isobutane, some methane, and oxidized carbon-species such as $CO_3^{2-}$, $HCO_3^-$ and $Na_2CO_3$ crystals. We used these experimental results to predict the stability of

immiscible isobutane fluid in equilibrium with silicate rocks with pelitic, mafic and ultramafic compositions. We also report theoretical calculations of equilibrium constants which predict that high pressure should stabilize aqueous hydrocarbons in the deep Earth. Altogether with a published experimental study at 600–700 °C and 1.5–2.5 GPa (ref. 14) and evidence from metasomatized ophicarbonates[6], our results suggest that immiscible hydrocarbon fluids might be stable at subduction-zone and upper mantle conditions in the presence of silicate-rock assemblages. Under these circumstances, hydrocarbon fluids might move as a separate phase from aqueous fluids or melts. These immiscible hydrocarbons could add a new way of transferring carbon within subduction-zone environments. Moreover, if these hydrocarbon fluids move into a shallower environment at pressures less than about 3.0 GPa, decomposition of the fluids into methane and carbon dioxide might result in metastable equilibria that could provide a source of methane in natural gas or food for a deep biosphere.

## Results

**Isobutane formation from sodium acetate**. We carried out an experimental study of the stability of aqueous acetate at high pressure. To specifically focus on the role of pressure, the experiments were carried out at a temperature in the range of previously published low-pressure experiments of acetate decomposition. Our experimental conditions of 3.0 GPa and 300 °C fall within the wide range of model temperatures and pressures[22] experienced by subducting slabs (Fig. 1). Indeed, it can be seen in Fig. 1 that our experimental conditions and those of ref. 14 showing experimental evidence of immiscible methane-rich fluids at 600–700 °C and 1.5–2.5 GPa bracket a large part of the expected range of subduction-zone pressure-temperature conditions. We studied 0.95 mol l$^{-1}$ $CH_3COONa$ solutions in six diamond anvil cell experiments at 300 °C and 2.4–3.5 GPa with a platinum or gold liner and experimental durations of 0.8–60 h (Supplementary Table 1 and Supplementary Fig. 1). Raman spectroscopy was the principal characterization method, together with analysis of the microscopic images obtained under *in situ* and quenched conditions.

We found that immiscible droplets of hydrocarbon fluid became clearly visible optically about two to three hours after reaching the peak temperature (Fig. 2a,b). The immiscible hydrocarbon was identified as mainly isobutane with minor amounts of ethane, propane and 2-methylpentane based on Raman spectra at high pressure-temperature conditions and after quenching (Fig. 2c,d, Supplementary Table 2 and Supplementary Fig. 2). The amount of fluid hydrocarbon was estimated by assuming the droplets were solely isobutane and counting the number of optically visible droplets (see Methods). The results are represented in Fig. 3a,b and Supplementary Table 3. We estimated that about 45% of the carbon in acetate was converted to isobutane. About 13% remained as aqueous acetate even after 60 h. Only about 11% of the initial carbon became methane, and the remaining 31% (by subtraction) became C(IV) species including aqueous $HCO_3^-$, $CO_3^{2-}$ and $Na_2CO_3$ crystals. No $H_2$ was detected in any of the experiments either at high pressures and temperatures or in the gas bubbles generated during quenching.

The evolution of the experimental system during the first ten hours is summarized in Fig. 3a. The immiscible droplets started forming at about 20 min (Exp_6, Au liner) to 2 to 3 h (Exps_1, 2, 3 and 5, Pt liner). After 2–3 h at 300 °C, some $Na_2CO_3$ crystals (Supplementary Fig. 3a) precipitated in Exp_1 to 3 but not in Exp_5 and 6. In Exp_4, which lasted for <50 min, the time was

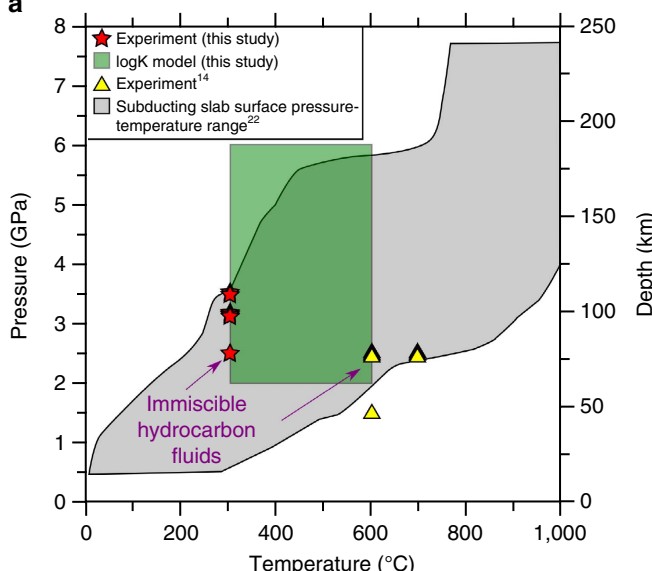

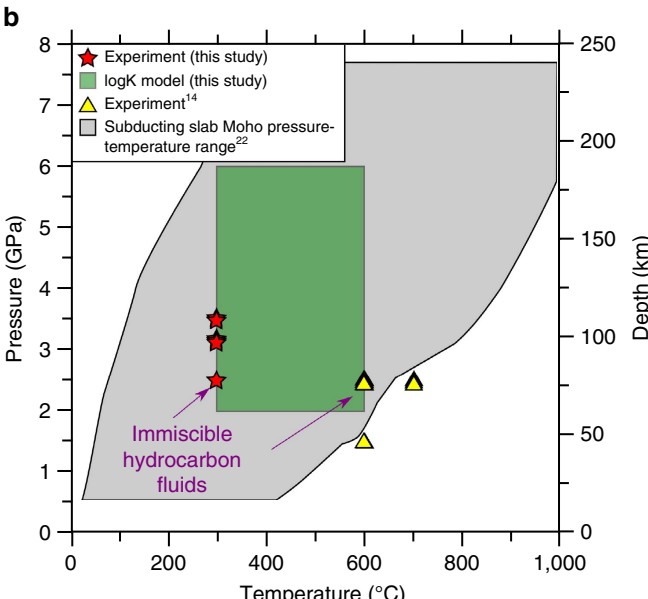

**Figure 1 | Comparison of experimental conditions involving immiscible hydrocarbon fluids those in subduction zones.** (**a**) The shaded grey zone represents the P–T range in W1300 model[22] of the surface of subducting slabs, and (**b**) the grey zone represents rocks at 7 km below the surface of subducting slabs. The red stars and yellow triangles represent the experimental conditions of this study and ref. 14, respectively, in which immiscible hydrocarbon fluids were formed. The green box represents a field of predominance of aqueous butane relative to acetate and methane suggested by calculated equilibrium constants for aqueous butane decomposition and acetate decarboxylation reactions.

too short to detect immiscible droplets or crystals (with Pt). However, a quenched gas bubble at the end of Exp_4 did reveal the presence of methane but no $H_2$. It can also be seen in Fig. 3a that bicarbonate was detected in the aqueous fluid after only one hour, and that the aqueous acetate to bicarbonate ratio dropped rapidly in the first four hours. These results indicate that the main initial reaction within the first one to two hours was decarboxylation of acetate according to equation (1).

$$CH_3COO^- + H_2O \rightarrow CH_4 + HCO_3^- \tag{1}$$

**Overall evolution of the experimental system.** During the overall long-term evolution of the experiments (Fig. 3b), a clear trend can be seen with increasing amounts of immiscible hydrocarbon reaching a plateau of nearly 50% of the total carbon in both gold (Exp_6) and platinum (Exp_3) runs; however, the amount of hydrocarbon increased earlier and faster in contact with gold. Furthermore, in Exp_3, part of the immiscible isobutane (8.6% of the total carbon) decomposed after the quench in temperature (quench_1) (Fig. 3b). However, the quench in pressure (quench_2) barely influenced the hydrocarbon content. After quench_1 there were no new species detected. Therefore, we assumed the decomposition reaction during quench_2 resulted from equation (2).

$$4C_4H_{10} + 9H_2O \rightarrow 13CH_4 + 3HCO_3^- + 3H^+ \tag{2}$$

According to this reaction, 7 out of 8.6% of carbon from the decomposed isobutane went into methane. The methane in the final gas bubbles after quench_2 in Exp_3 accounts for $\sim$11% of the total carbon in the system (Supplementary Fig. 4). This implies that 70% of the total methane comes from the decomposition of isobutane, and the remaining 30% formed at high pressure and temperature from the decarboxylation of acetate (equation (1)).

It can also be seen in Fig. 3b that the ratios of the $CH_3COO^-$/ $HCO_3^-$ peak areas decreased from about 3.7–4.5 to 0.2–0.3 and stayed constant after about 20 h, indicating that the aqueous system had reached a steady state. Differences between the $CH_3COO^-$/$HCO_3^-$ ratios in Exp_1 and 2 before the steady state was reached may reflect differences in the pressure-temperature paths of Exp_1 and Exp_2 (Supplementary Fig. 1), and the slightly higher average pressure in Exp_2 (3.4 GPa) compared with Exp_1 (3.1 GPa). A peak suspected to be the C–C stretching of aqueous isobutane was present from 1 to 2.5 h, but disappeared by 4.7 h, probably indicating that the aqueous solution became supersaturated[23] with isobutane before it separated as an immiscible fluid (Fig. 4). Aqueous carbonate ions became visible in the Raman spectra after $\sim$19 h (Fig. 4). The peak areas of $HCO_3^-$ and $CO_3^{2-}$ at the end of the experiment at 25 °C yield a final pH of 9.2 (initial 9.4) based on previous experimental calibrations[24]. Whether the steady state represents equilibrium or not remains to be established. Nevertheless, there was measurable acetate remaining in the system even after the steady state was reached.

**The carbon budget.** The estimated overall carbon budget after quenching is shown in Fig. 5a. The composition of dissolved gases detected by quenching and volatile formation is shown in Fig. 5b. On the basis of the carbon and sodium mass balances, the charge balance and the known $HCO_3^-$/$CO_3^{2-}$, we further divided the C(IV) budget into $Na_2CO_3$ crystals (35%), $HCO_3^-$ (60%) and $CO_3^{2-}$ (5%), as indicated in Fig. 5c (see Methods). We note that our detection of an abundant immiscible, hydrocarbon heavier than methane coexisting with water differs drastically from the predictions of previous widely-used COH models of upper mantle fluids[15] and previous experimental results[8–9]. In particular, the immiscible hydrocarbon that formed in our experiments is so abundant that it contains much more of the system carbon than there is in the methane that formed.

**Discussion**

Previous experimental studies of acetic acid ($CH_3COOH$) decomposition at 300–400 °C and low pressure[25,26] obtained complex hydrocarbon oils. The latter were not formed from Na-acetate solutions, and the main products were $CH_4$ and $CO_2$. It was suggested that acetic acid solutions decomposed to

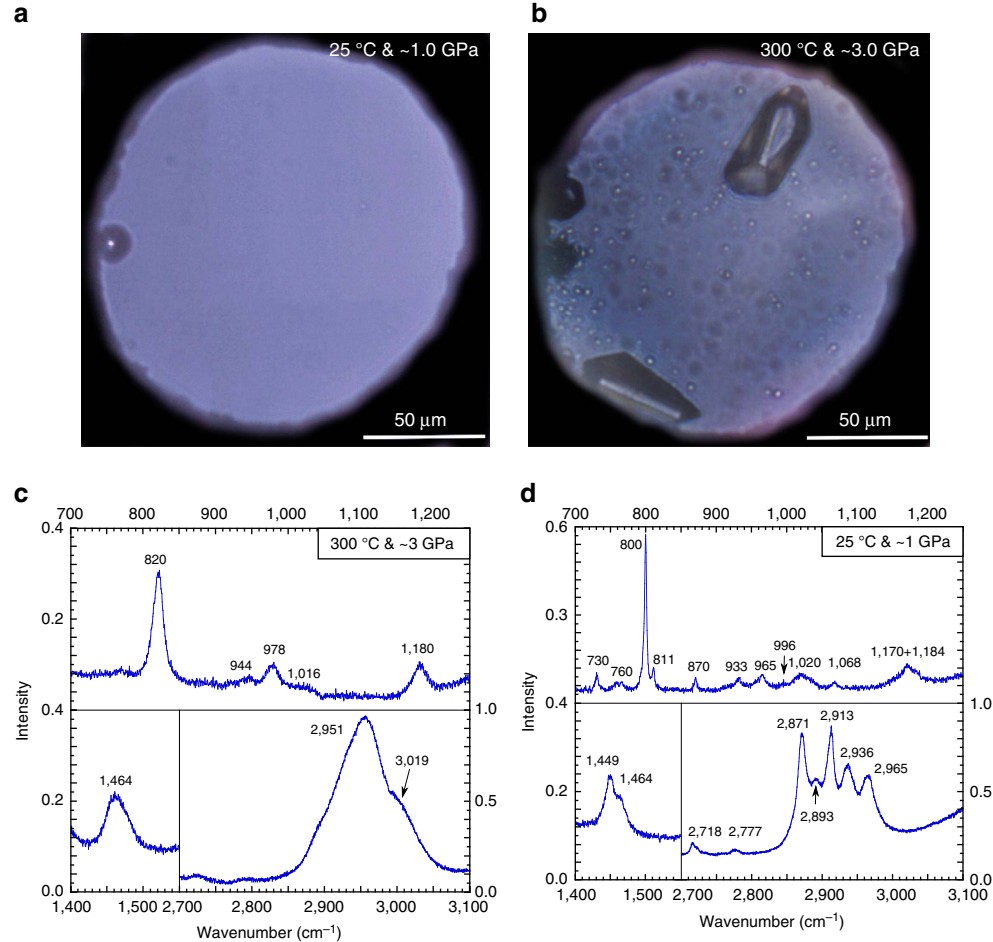

**Figure 2 | Optical views of the diamond anvil cell and Raman spectra of the immiscible hydrocarbon droplets of Exp_3.** (**a**) The clear starting aqueous fluid; (**b**) Aqueous fluid, droplets of hydrocarbon fluid & crystals after 66.9 h at 300 °C. Both have a scale bar of 50 μm. (**c**) Raman spectrum of a droplet of hydrocarbon fluid at 300 °C and ~3 GPa; (**d**) Raman spectrum of a droplet of hydrocarbon liquid at 25 °C and ~1 GPa (after quench_1).

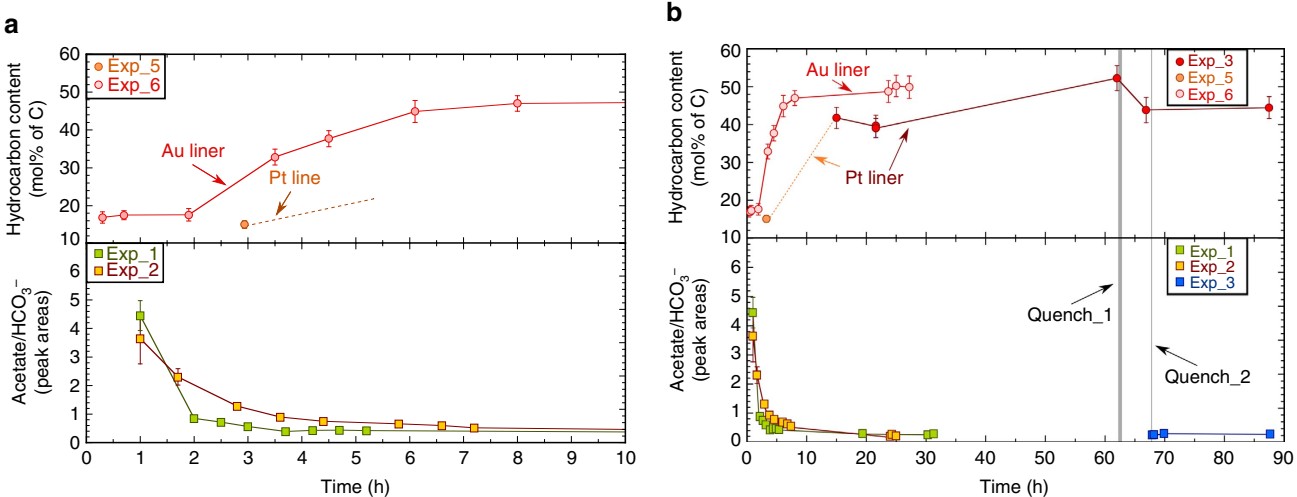

**Figure 3 | The evolution of the experimental system with time.** (**a**) The first 10 h; (**b**) Up to 87 h. Calculations of error bars can be found in methods. Two sets of quenching were performed in Exp_3: (1) quench_1—the system temperature took 1.5 h to decrease from 300 to 25 °C and pressure from 3 to ~1 GPa after 62.2 h heating; (2) quench_2—the system pressure took 0.1 h to decrease further to 4.24 MPa after 68.5 h.

hydrocarbons through polymerization reactions via the formation of acetone and then isobutene at 450–500 °C (ref. 27), followed by reduction to isobutane. However, a subsequent experimental study[20] suggested that in these acetic acid experiments, FTT reactions played a significant role. In contrast with the low-pressure experimental results, our experiments with Na-acetate

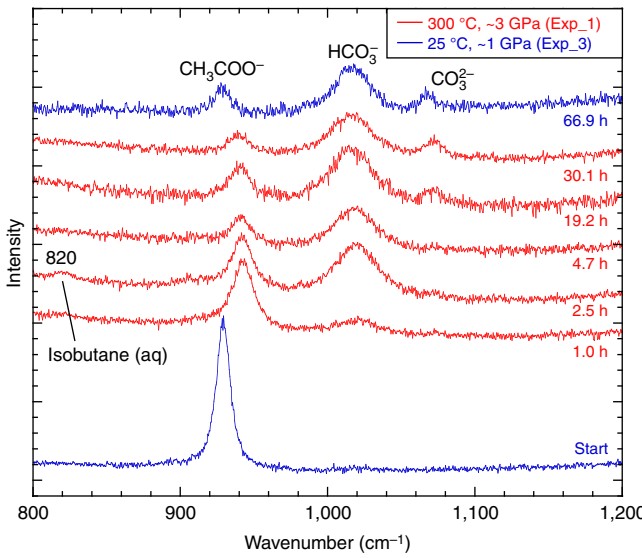

**Figure 4 | Raman spectra of the aqueous species through time.** The acetate peaks shift to higher wavenumbers at higher pressure. A broad peak at 820 cm$^{-1}$ is apparent between 1.0 and 2.5 h and disappeared by 4.7 h (possibly aqueous isobutane, see text).

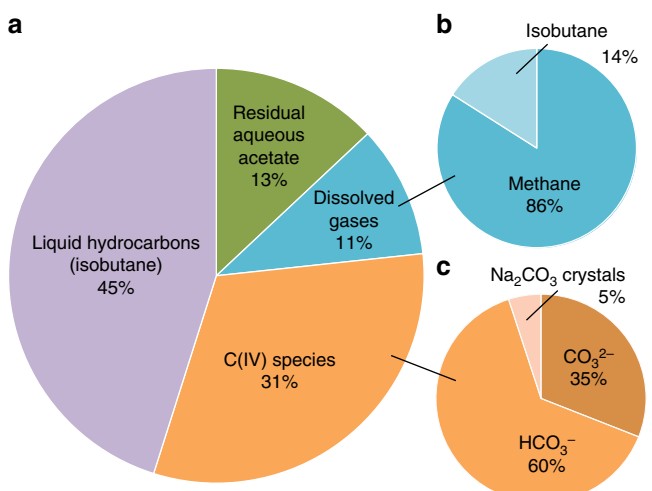

**Figure 5 | The carbon budget of the quenched system.** The condition is 25 °C and 4.2 MPa, and numbers are by mol% total C. For example, 1 mole of isobutane has 4 moles of C but 1 mole of methane only has 1 mole. (**a**) The overall budget of carbon species; (**b**) The budget of gases exsolved after quenching; (**c**) The distribution of C(IV) species. The amounts of gases and hydrocarbons were estimated after quench_2 (Fig. 3). Residual acetate was quantified from the Raman spectrum after quench_1 (see Methods).

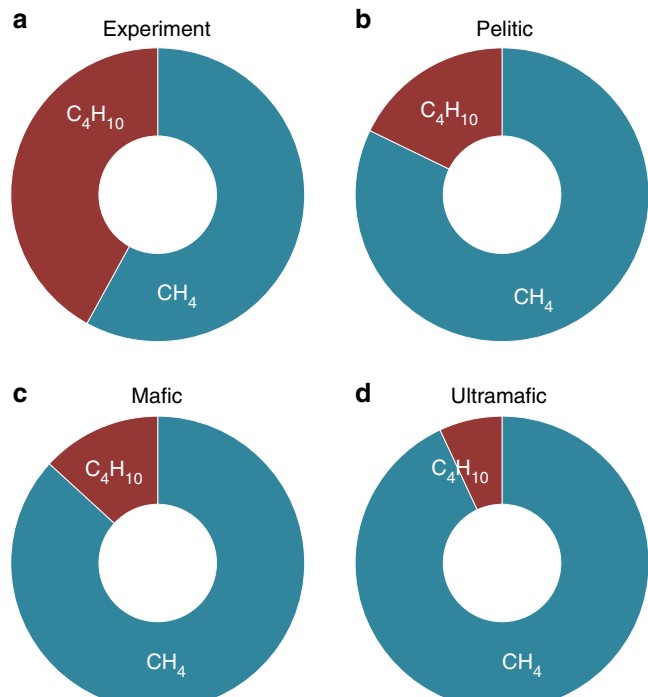

**Figure 6 | The compositions of immiscible hydrocarbon fluids from experiments and models.** (**a**) the experiment; (**b**) the model starting with a pelitic-rock composition, ending with pH = 4.6, logfo$_2$ = −32.3 and 22 mmol hydrocarbons; (**c**) the model starting with a mafic-rock composition, ending with pH = 4.7, logfo$_2$ = −32.5 and 33 mmol hydrocarbons; (**d**) the model starting with a ultramafic-rock composition, ending with pH = 5.0, logfo$_2$ = −32.7 and 45 mmol hydrocarbons. The ratios are by mol% of hydrocarbon species. At equilibrium, the hydrocarbon fluid in the pelitic system has more isobutane relative to CH$_4$, but in the ultramafic system there is more hydrocarbon fluid overall.

can be represented by equation (3).

$$13CH_3COO^- + Na^+ + 4H_2O$$
$$\rightarrow 4C_4H_{10}(l) + 6HCO_3^- + 3.5CO_3^{2-} + 0.5Na_2CO_3(s) + H^+$$
$$(3)$$

This reaction stoichiometry agrees with the independently quantified amounts of hydrocarbons and C(IV) species (see Methods).

Our experimental results clearly indicate a special role for high pressure in facilitating fluid hydrocarbon synthesis and immiscibility in the presence of water. Altogether with the recent experimental evidence of immiscible methane-rich fluids at 600–700 °C and 1.5–2.5 GPa (ref. 14) our results point to the possible importance of immiscible hydrocarbon fluids under conditions that bracket a large part of the expected range of subduction-zone pressures and temperatures (Fig. 1). It should be emphasized however that our experimental study and that of ref. 14 started with simple carboxylic acid solutions. The effect of natural silicate rock assemblages at high pressures remains to be established. To investigate these compositional effects, we developed a theoretical model at 300 °C and 3.0 GPa with 0.1 mol per kg H$_2$O acetate solution coexisting with silicate mineral assemblages (Supplementary Table 4).

We first calibrated the free energies of the fluid isobutane and methane, and that of the Na$_2$CO$_3$ crystals at 300 °C and 3.0 GPa, using our experimental results. We then created a model for acetate solutions in the presence of pelitic, mafic and ultramafic mineral assemblages at 300 °C and 3.0 GPa. The model represents

produced abundant isobutane. Our results are not consistent with FTT synthesis for the following reasons: (1) the amounts of products of FTT synthesis show a logarithmically decreasing trend with carbon number, whereas we found much higher amounts of C$_1$ and C$_4$ compared to C$_2$ and C$_3$ (C$_n$ refers to the hydrocarbon species with n carbons); (2) H$_2$ is a key reactant in FTT reactions, but we did not detect any H$_2$; (3) FTT reactions in experiments may often take place in a separate gas phase, but a gas phase was absent in our experiments. Thus, the FTT mechanism is not appropriate for our system. We suggest that the overall hydrocarbon formation reaction in our experiments

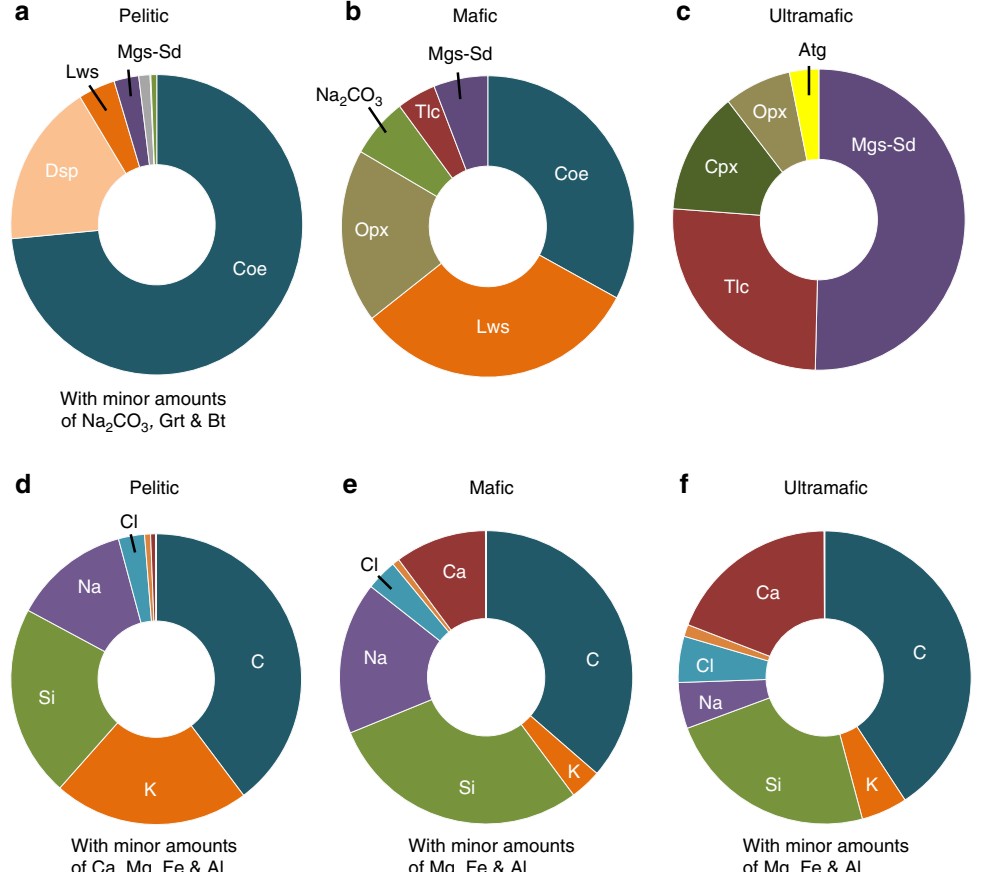

**Figure 7 | Three types of mineral assemblages and corresponding aqueous fluid compositions in equilibrium with immiscible hydrocarbon fluid.**
The mineral assemblages at the end of the (**a**) pelitic, (**b**) mafic and (**c**) ultramafic models. The aqueous fluid compositions at the end of the (**d**) pelitic,
(**e**) mafic and (**f**) ultramafic models. Mineral abbreviations: Atg, Antigorite; Bt, Biotite; Coe, Coesite; Cpx, Clinopyroxene; Dsp, Diaspore; Grt, Garnet; Lws,
Lawsonite; Mgs-Sd, Magnesite-Siderite solid solution; $Na_2CO_3$, Sodium Carbonate; Opx, Orthopyroxene; Tlc, Talc. The aqueous fluids are rich in C, Si, Na,
Si, Ca and Cl species.

a Gibbs free energy minimization of the acetate solutions in the presence of each bulk chemical composition. The results shown in Figs 6 and 7 indicate that abundant fluid hydrocarbons (methane and isobutane) are predicted to coexist in equilibrium with model silicate rocks representative of pelitic, mafic and ultramafic bulk compositions. The hydrocarbons contain about 10–25% of the total carbon in the system. The final $fo_2$ values for the three different bulk compositions are also geologically reasonable as they are about 1–2 log units less than the quartz-fayalite-magnetite (QFM) buffer.

To further study the effects of higher temperatures and pressures, we used the Deep Earth Water model[28] to predict the equilibrium constants of isobutane decomposition and acetate decarboxylation reactions from 200 to 600 °C and 2 GPa to 6 GPa. The predicted equilibrium constants for the decarboxylation of acetate (equation (1)) show that pressure strongly opposes decarboxylation, even up to 600 °C (Supplementary Fig. 5a), consistent with a recent study[29]. Similar trends are apparent for the decomposition of aqueous isobutane to methane and bicarbonate (Supplementary Fig. 5b), suggesting that once formed, immiscible hydrocarbons and water might be stable in subduction-zones at high pressures.

To conclude, high pressure and water can potentially stabilize immiscible hydrocarbon fluids coexisting with aqueous fluids containing organic species in the deep Earth. If correct, these hydrocarbon fluids and aqueous organic species could be an important part of Earth's deep carbon cycle[3,5], facilitating the migration of carbon in a variety of fluids[30], and possibly the formation of diamonds[31,32] during mantle metasomatism. If such fluids ascend into the crust at lower pressures, they may decompose to methane, carbonate, and other aqueous organics in metastable equilibria[33], supplying energy to a deep biosphere[34]. Advances in the study of clumped isotopes have determined formation temperatures of ~150 to ~220 °C for abiogenic (thermogenic) methane found in various geologic environments[35,36], a similar temperature range to our quenching temperatures for the decomposition of immiscible isobutane. Furthermore, recent analyses of marine sediments[34] have revealed the existence of microbial communities at ~1.5–2.5 km below the seafloor. Presumably such organisms could take advantage of the energy and carbon sources provided by the decomposition of deep hydrocarbons.

## Methods

**Experimental details.** A set of speciation experiments was conducted with a $CH_3COONa$ solution at 300 °C and 2.4–3.5 GPa using a membrane-type diamond anvil cell[37], mounted with ultra-low fluorescence anvils with a 500 μm-diameter culet. The starting fluid was prepared by mixing 5.00 ml of 1.0 mol l$^{-1}$ NaOH (Merck KGaA) and 0.286 ml of pure $CH_3COOH$ (Merck KGaA, 100% purity, density = 1.05 kg l$^{-1}$) solutions. The resulting volume was 5.286 ml of $CH_3COONa$ solution with a concentration of 0.95 mol l$^{-1}$ and an initial pH of 9.4 according to reaction (4).

$$NaOH + CH_3COOH \rightarrow Na^+ + CH_3COO^- + H_2O \qquad (4)$$

The solution was loaded into a reaction chamber with a diameter of 150 μm centred in a pre-indented 100-μm-thick stainless steel gasket. A 200-μm hole

was first drilled, then filled with Pt or Au, and drilled again into 150 µm. The 50-µm-thick Pt or Au wall prevented the reaction between the solution and the stainless-steel gasket. To avoid organic contamination, the gaskets were sonicated in a high concentration $H_2O_2$ solution for at least 5 min. The diamond culets, tweezers and needles used for loading the cell were also cleaned using cotton swabs dipped with $H_2O_2$ solution. The comparison of the Raman spectrum of the initial fluid and the $CH_3COONa$ solution confirmed no detectable contamination from other carbon species. One to three ruby spheres were added to monitor the pressure during the experiment[38]. The diamond anvil cell was heated externally using a resistance-heating coil. The temperature was measured through a K-type thermocouple touching one diamond anvil, and controlled automatically to keep the temperature constant within 1 °C. The pressure inside the reaction chamber was increased and regulated by pumping Helium gas into the cell membrane, using an automatic pressure regulator (Sanchez technologies). The pressure was measured using the shift of the R1 peak of the ruby sphere[39], the temperature dependence of which has been calibrated for the present batch of ruby spheres[40].

Raman spectra were collected on a confocal LabRam HR800 spectrometer (Horiba Jobin–Yvon) of 800 mm focal length equipped with a 532 nm Nd:YAG laser (Torus Laser, Laser Quantum) and a Mitutoyo × 50 long working distance objective (0.42 N.A.). The laser power at the sample was always < 10 mW. Each Raman spectrum was acquired between 200 cm$^{-1}$ and 4,300 cm$^{-1}$ except the range with strong diamond peaks, about 1,300–1,700, 2,000–2,700 cm$^{-1}$ with a spectral resolution of 0.3 cm$^{-1}$. The Raman spectra were processed using the PeakFit software; the baseline was subtracted and peak parameters determined assuming a Voigt shape.

**Carbon budget.** The total number of moles of C in the system was calculated through equation (5) based on the known concentration in the initial aqueous solution and the volume of the cell:

$$C_{tot} = 2 \times 0.95 \, mol \, l^{-1} \times 100 \, \mu m \times \left(\frac{150}{2} \mu m\right)^2 \times \pi = 3.4 \times 10^{-9} \, mol \quad (5)$$

If not noted, all the percentages used below are in moles of carbon, not moles of the molecules. For example, 1 mole of isobutane has 4 moles of C, but 1 mole of methane only has 1 mole of C.

We quantified the amount of gases using the ideal gas law. The pressure of the gas bubble was calibrated based on the ratios of the $v_3$ and $2v_2$ peak areas of methane[41]. The average pressure $P$ equals $4.24 \pm 1.3$ MPa based on four measurements of Exp_3. The total volume of the three bubbles (Supplementary Fig. 4) were estimated based on several geometrical assumptions: (1) the top bubble has a cylindrical shape because it is big enough to touch the culets of the diamond anvils; (2) the two bottom bubbles have a spherical cone-shape. The integrated volume from the pictures was in units of pixels$^3$.

The volumes of the two types of gas bubble are calculated as: (1) the ratio of the volume of the cylindrical top bubble to the volume of the whole cell can be expressed as the ratio of the area of the bubble to the cell (56,486/559,256 = 9.95%) (Supplementary Fig. 4), so the volume of the top bubble is 175,746 µm$^3$. (2) The estimated volume of the two bottom gas bubbles in pixels$^3$ was converted into µm$^3$. By scaling the area of the cell in the picture (567,971 pixel$^2$) to the surface of the compression chamber (17,671 µm$^2$), we obtained a conversion factor of 5.669 (pixel per µm). The volume of the two bottom gas bubbles is 4,526,113 pixel$^3$, that is, 24,843 µm$^3$. So the total volume of gas bubbles is 200,589 µm$^3$. Then we used the ideal gas law (equation (6)) to calculate the amount of gas:

$$PV = nRT \quad (6)$$

where $T = 298.15$ K (25 °C), $R = 8.314$ J mol$^{-1}$ K$^{-1}$, $P = 4.24$ MPa, $V = 200,589$ µm$^3$. Thus, the amount of gas $n$ is $0.34 \times 10^{-9}$ mol. On the basis of the relative peak area of the C–H stretching of isobutane and methane (Supplementary Fig. 3b), we know that methane is 95.5% and isobutane is 4.5%. Ethane and propane are < 1%. Ignoring the ethane and propane, we obtain the number of moles of carbon in the gas phase to be $(0.34 \times 0.955 + 0.34 \times 0.045 \times 4) \times 10^{-9} = 0.386 \times 10^{-9}$ mol, which represents 11% of total carbon.

We quantified the amount of the hydrocarbon fluids by counting the number of optically visible droplets, knowing the volume of the cell, and estimating the density of isobutane at pressure and temperature. It is assumed that the droplets are distributed in the system evenly. That is, in each optical picture, the ratio of the areas of the droplets to the cell was assumed to represent the volume of the hydrocarbons to that of the cell. However, when examining the pictures with droplets, we found that some of the in-focus droplets were clearer than the out-of-focus ones. Therefore, we counted the most apparent droplets as a lower limit ($a$) of the hydrocarbon amount, and all visible droplets as the higher limit ($b$). The values plotted in Fig. 3 are $(a+b)/2$ and error bar as $(b-a)/2$. In this way, we obtained the volume of the droplets and uncertainties. Using an equation of state of fluid isobutane[42], we calculated the density of isobutane within our experimental pressure-temperature range, with the following results: (1) 0.558 kg l$^{-1}$ at 25 °C and 4.24 MPa, (2) 0.839 kg l$^{-1}$ at 25 °C and 1.0 GPa and (3) 0.923 kg l$^{-1}$ at 300 °C and 3.0 GPa. Using the volume and density, we obtained the mass of isobutane (assuming the droplet is pure isobutane).

For instance, at 300 °C and 3.0 GPa, the density of pure water is 1.33 kg l$^{-1}$. We can calculate the volume of the water fluid to be $1.33 \times 10^6$ µm$^3$, smaller than initial

volume due to compression. In the last picture taken before quench_2 of Exp_3, we know that the area of hydrocarbons is about $2.11 \pm 0.13\%$ of the total area of cell, so we can calculate the mass of hydrocarbon droplet (isobutane) to be $m = 2.11/100 \times 1.33 \times 10^6 \, \mu m^3 \times 0.923 \, kg \, l^{-1} = 2.59 \times 10^{-8}$ g, that is, $4.456 \times 10^{-10}$ mol of isobutane. Multiplying the value by 4 gives a value of $1.78 \times 10^{-9}$ mole carbon, representing 52% of the total carbon ($3.4 \times 10^{-9}$ mole). The rest of the estimated carbon contents of hydrocarbons are listed in Supplementary Table 3.

We quantified the amount of residual aqueous acetate by directly converting the peak areas into concentrations based on a standard solution. The pressure and temperature ($\sim 1$ GPa and 25 °C) at which the Raman spectra of residual aqueous acetate was taken was similar to the pressure and temperature conditions at which the standard 0.95 mol l$^{-1}$ $CH_3COONa$ solution was measured before heating. Furthermore, the laser power and the focus length of two measurements were also identical. Therefore, we directly converted the ratio of the peak areas of acetate into ratio of concentrations. The initial peak area was 17,209.44 and the final peak area was 2,270.84, corresponding to a concentration of 0.125 mol l$^{-1}$, 13% of the starting 0.95 mol l$^{-1}$.

We quantified the oxidized carbon species by mass balance and charge balance. The total C(IV) species is about 31% of the total carbon. For simplicity, we assumed the initial sodium acetate was 1,000 mmol, so there were initially 2,000 mmol of total carbon in the system, resulting in 130 mmol acetate and 620 mmol C(IV) species. The carbon budget of C(IV) species is divided based on equations (7)–(10).

$$[CO_3^{2-}] + [HCO_3^-] + [Na_2CO_3] = 620 \, mmol - \text{Carbon balance} \quad (7)$$

$$[Na^+] + 2[Na_2CO_3] = 1,000 \, mmol - \text{Sodium balance} \quad (8)$$

$$\frac{[HCO_3^-]}{[CO_3^{2-}]} = 12.25 - \text{Converted from ratio of peak areas} \quad (9)$$

$$2[CO_3^{2-}] + [HCO_3^-] + [CH_3COO^-](\text{known}) = [Na^+] - \text{Charge balance.} \quad (10)$$

Combining equations (7)–(10), we obtained the detailed C(IV) budget to be 220 mmol $Na_2CO_3$ crystals (35%), 370 mmol $HCO_3^-$ (60%) and 30 mmol $CO_3^{2-}$ (5%).

The mass balance was calculated based on moles of carbon instead of moles of molecules. Thus, if we assume the initial acetate is 1,000 mmol (that is, total C is 2,000 mmol), 45% of the C (900 mmol) in fluid hydrocarbon only gives 225 mmol of isobutane, 620 mmol of C(IV) species and 220 mmol of methane. Approximately 870 mmol of acetate decomposed during the experiment. According to the overall reactions (equations (1) and (3)), we calculated that 650 mmol acetate reacted in equation (1) and 220 mmol from equation (2), assuming that 220 mmol of methane comes from equation (2). The two reactions give a result of 220 mmol methane, 720 mmol C(IV) and 200 mmol isobutane, which is consistent with our experimental values. The small remaining imbalance in carbon valence is within the uncertainties of the measurements and estimations.

**Data availability.** The data that support the findings of this study are available within the paper and its Supplementary Information Files. Raw data, that is, diamond anvil cell photos, are available from the corresponding author on reasonable request.

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

## Acknowledgements

This research was supported by grants from the Sloan Foundation through the Deep Carbon Observatory (Reservoirs and Fluxes and Extreme Physics and Chemistry programs). We are grateful for the help and support of the Johns Hopkins University, the Geophysical Laboratory of the Carnegie Institution of Washington, and the Ens de Lyon. The Raman facility at the Ens de Lyon is supported by the Institut National des Sciences de l'Univers (INSU). The department of chemistry of the Ens de Lyon provided a comprehensive series of standard organic liquids, which enabled measurements of high-resolution Raman spectra that were decisive in the interpretation of the complex *in situ* system. We thank A. Vitale Brovarone for kindly sharing his results and thoughts. We acknowledge helpful discussions with Y. Fei, C.E. Manning, C. He, J. Hao, H. Hu, C.M. Schiffries, E.L. Shock, R.M. Hazen, C. Glein, M. Galvez, and T. McCollom. F. Huang acknowledges X. Wang for help in life in Lyon. D.A. Sverjensky thanks W. Link for his invaluable encouragement and advice on science and life.

## Author contributions

F.H., I.D. and D.A.S. initiated the study. F.H., I.D., H.C. and G.M. carried out the experiments. F.H., I.D. and D.A.S. performed the experimental data analysis and collaborated in writing the manuscript. All authors contributed to the discussion of the results and revision of the manuscript.

## Additional information

**Competing interests:** The authors declare no competing financial interests.

