## [Peer Review File · Nature Communications]

Reviewers' Comments:

Reviewer #1 (Remarks to the Author)

1. Summary

Stability experiments of acetate solutions were carried out at 300 C and 2.4-3.5 GPa. Through Raman characterisation, it was found that CH₄ is the initial hydrocarbon formed through decarboxylation of acetate, followed by isobutane formation. After quenching the pressure, isobutane was found to decompose to methane and bicarbonate.

The manuscript is well written and easy to read, and the figures are of high quality. I have no objections to the methodology or results of the experiments, which are all well documented. However, I am not convinced of how applicable the results are to the study of the deep carbon cycle. I have outlined those concerns below.

2. Experimental conditions for the formation of isobutane from acetate

The acetate stability experiments were carried out 300 C and around 3 GPa. The authors state that they purposefully used the same temperatures as previous low-pressure experiments of acetate stability.

While these conditions are a useful starting point to establish the role of pressure, the study could have been extended to evaluate acetate stability and its potential role in deep hydrocarbon formation at conditions that are geologically relevant. As far as I'm aware there are no geological settings where 300 C and 3 GPa is typical. As is, this is an interesting theoretical study on the role of pressure in hydrocarbon formation from acetate, but it has no direct applicability to studies of natural samples.

In Lines 144-145 the authors state that the importance of their study is that experimentally-produced isobutane might be stable in subduction zones and therefore is important for our understanding of the deep carbon cycle. However, they have provided no evidence of this.

The authors themselves state in line 137-138 that they cannot establish whether this process is applicable in subduction zone settings. They cannot go on to claim that the experimentally-formed isobutane will percolate upwards into the mantle wedge and crust and decompose into methane, if they haven't proven that isobutane forms in the subducting slab in the first place.

The authors could consider extending their experimental work to natural conditions in order to have a more complete dataset that could then be applied to the study of the deep carbon cycle on Earth.

For these reasons, I cannot recommend to the journal that this study be accepted for publication. I suggest that the authors consider extending their experiments, and then resubmit for further evaluation.

3. Acetate in natural samples

Can the authors provide an example of acetate occurrence in natural samples that could be subducted? Or is this simply a theoretical pathway?

4. Existence of hydrocarbons heavier than methane

In lines 21-22 and lines 37 -38 the authors claim that the evidence for heavier hydrocarbons in the deep crust and mantle is 'meager'.

The following studies could be given as counter examples along with this statement, although I am sure there is more evidence in the literature:

- a) Hirai et al., 2009 (Physics of the Earth and Planetary Interiors) showed ethane formation from methane at $P > 10$ GPa
- b) Kolesnikov et al., 2009 (Nature Geoscience) showed that heavier hydrocarbons can co-exist with methane in the lithosphere ($P > 2$ GPa)
- c) Lobanov et al., 2013 (Nature Communications) showed that at pressures typical of the lower mantle ($P > 24$ GPa) reduced carbon-bearing fluids are rich in heavier hydrocarbons instead of methane.

5. Minor comments

Lines 16-18 - lithospheric diamonds (depths of at least 100 km and temperatures of 1100 C) with fluid inclusions of abiogenic methane have recently been documented (Smit et al., 2016 - Lithos).

Line 27 - should 'decarboxylation of isobutane' read 'decarboxylation of acetate' instead?

Reviewer #2 (Remarks to the Author)

Vinegar into oil: High pressure changes aqueous acetate to immiscible hydrocarbon fluid. By Huang et al.

Overall, a very interesting paper with some excellent experimental data. The authors drift a little in their message and need to define an objective, choose a narrative and then maintain it from start to finish.

Title: The title prepares the reader for a chemical engineering article but the introduction sets a very different scene. I would remove the "Vinegar into oil" part and add a second title that was something like "persistent organic fluids in the subsurface"

Summary Paragraph Page 2: I think "decarboxylation of isobutene" should be "decarboxylation of acetic acid" or "decarboxylation of acetate".

Main text Page 2: "subducted solid organic matter ..." is slightly incorrect because solid organic matter subjected to increasingly high pressures and temperatures converts to liquid and then gas. This is the concept behind production ratios. Why not say just "subducted organic matter ..."

Main text Page 3: the observation of immiscible hydrocarbons must indicate that the saturation of hydrocarbons in the water is exceeded. This point is made on page 5, but then apparently forgotten on page 6 when considering mantle models. At 300 °C and high pressure water acts as an organic solvent and can dissolve hydrocarbons (e.g. Luong D., Sephton M.A. and Watson, J.S. 2015. Subcritical water extraction of organic matter from sedimentary rocks, *Analytica Chimica Acta*, 879, 48-57. doi: 10.1016/j.aca.2015.04.027.)

Main text Page 7: "... pressure opposes decarboxylation ..." would be supported by the hint of a mechanisms. Presumably hydrogen bonding is involved as implied for recently published data on polymers (e.g. Montgomery W, Potiszil C., Watson J.S. and Sephton M.A. Sporopollenin, a natural copolymer, is robust under high hydrostatic pressure. *Macromolecular Chemistry and Physics*, in press. doi: 10.1002/macp.201600142)

Reviewer #1 (Remarks to the Author):

1. Summary

The manuscript is well written and easy to read, and the figures are of high quality. I have no objections to the methodology or results of the experiments, which are all well documented. However, I am not convinced of how applicable the results are to the study of the deep carbon cycle. I have outlined those concerns below.

We will reply to these concerns one by one.

2. Experimental conditions for the formation of isobutane from acetate

The acetate stability experiments were carried out 300 C and around 3 GPa. The authors state that they purposefully used the same temperatures as previous low-pressure experiments of acetate stability. While these conditions are a useful starting point to establish the role of pressure, the study could have been extended to evaluate acetate stability and its potential role in deep hydrocarbon formation at conditions that are geologically relevant.

We emphasize in the revised text (lines 62 – 66, and 145 – 150) that our experimental pressures and temperatures are indeed relevant to subduction zones. In order to emphasize this point, we have included a new Supplementary Figure 1 that shows the expected ranges of P-T conditions for subduction zone models (Syracuse et al., 2010). It can be seen in this figure that our experimental conditions and those of Li (2017) bracket a large part of the expected P-T range for subduction zones. Furthermore, the relevance of our experimental results is supported by theoretical model calculations of equilibrium constants for isobutane and acetate decomposition to higher temperatures and pressures (lines 165 – 172). We also describe below theoretical modeling that extend our experimental results to geologically interesting bulk compositions relevant to subduction zones (lines 151– 164).

As far as I'm aware there are no geological settings where 300 C and 3 GPa is typical.

We show in Supplementary Fig. 1 that our P-T conditions do fall within the range predicted for subduction zones by Syracuse et al. (2010) and also revisited that (line 62 – 66 and 145 -150).

As is, this is an interesting theoretical study on the role of pressure in hydrocarbon formation from acetate, but it has no direct applicability to studies of natural samples.

We disagree. First of all, our study is experimental and theoretical, not just theoretical. Second, we cite in the revised text the recent experimental evidence by Li (2017) of immiscible hydrocarbon fluids at even higher temperatures than our study (lines 18- 19 and 37 – 40). Third, we cite in the revised text the recent fluid inclusion evidence by Brovarone et al. (Nat. Comm., in press) of high- pressure methane-rich hydrocarbon fluids in natural metasomatized serpentinites from the Alps (lines 20 – 21, 33 – 34). These results strongly support our suggestion that our study may help explain the occurrence of methane in fluid inclusions from the breakdown of isobutane at lower pressures than our study.

Finally, we also demonstrate the geologic applicability of our results by developing a model calibrated with our experimental data and applied to geologically relevant bulk compositions (lines 151 – 164), i.e. pelitic, mafic and ultramafic mineral assemblages at 300 °C and 3.0 GPa. The results showed that fluid isobutane could form in significant amounts with all rock types and different acetate concentrations (Fig. 5, Supplementary Fig. 7 and Table 4).

In Lines 144-145 the authors state that the importance of their study is that experimentally produced isobutane might be stable in subduction zones and therefore is important for our understanding of the deep carbon cycle. However, they have provided no evidence of this. The authors themselves state in line 137-138 that they cannot establish whether this process is applicable in subduction zone settings.

We have revised the manuscript (lines 144 to 175) to indicate that our experimental results and models, together with recent experimental results from Li (2017) and natural fluid inclusions from Brovarone et al. (2016), strongly suggest that immiscible fluid isobutane and methane can play an important role in the deep carbon cycle.

They cannot go on to claim that the experimentally formed isobutane will percolate upwards into the mantle wedge and crust and decompose into methane, if they haven't proven that isobutane forms in the subducting slab in the first place.

We do not claim that isobutane will migrate into the mantle wedge. Instead we suggest that any migration of isobutane to lower pressures from the slab would produce methane (see above).

The authors could consider extending their experimental work to natural conditions in order to have a more complete dataset that could then be applied to the study of the deep carbon cycle on Earth.

We agree that more experimental data is always desirable. However, the results of our current experiments are so different from expectations at lower pressures based on any previous experiments that they warrant publication. Furthermore, as indicated above, when our results are considered together with the recent experimental results from Li (2017) and natural fluid inclusions from Brovarone et al. (2016), the results strongly suggest that immiscible fluid hydrocarbons can play an important role in the deep carbon cycle.

For these reasons, I cannot recommend to the journal that this study be accepted for publication. I suggest that the authors consider extending their experiments, and then resubmit for further evaluation.

In our revised manuscript, we have shown that our experimental pressures and temperatures are consistent with models of subduction zones as described by Syracuse et al. (2010) and are also consistent with other recent studies of fluid inclusions synthesized experimentally and observed in natural samples as described above. And, as an additional response to the reviewer, we have expanded the theoretical modeling to help emphasize the possible geologic applicability of our study.

3. Acetate in natural samples

Can the authors provide an example of acetate occurrence in natural samples that could be subducted? Or is this simply a theoretical pathway?

Several studies show that the concentration of aqueous acetate can reach up to 1.0 mol/L in marine sediments (Wang & Lee, 1993; Henrichs, 1992) and up to 0.1 mol/L in sedimentary basins (Shock, 1988). This part of aqueous acetate is very likely to be subducted into deep Earth, and provide a source for hydrocarbon formation as demonstrated in this study. Furthermore, acetate is a convenient source of zero oxidation state carbon. Our theoretical models show that equilibration of acetate or a mixture of methane and carbon dioxide with silicate rock minerals

produces exactly the same results. This is to be expected, as thermodynamics is a function of temperature, pressure, and bulk composition.

4. Existence of hydrocarbons heavier than methane

In lines 21 22 and lines 37 38 the authors claim that the evidence for heavier hydrocarbons in the deep crust and mantle is 'meager'.

The following studies could be given as counter examples along with this statement, although I am sure there is more evidence in the literature:

a) Hirai et al., 2009 (Physics of the Earth and Planetary Interiors) showed ethane formation from methane at $P > 10$ GPa

b) Kolesnikov et al., 2009 (Nature Geoscience) showed that heavier hydrocarbons can coexist with methane in the lithosphere ($P > 2$ GPa)

c) Lobanov et al., 2013 (Nature Communications) showed that at pressures typical of the lower mantle ($P > 24$ GPa) reduced carbon bearing fluids are rich in heavier hydrocarbons instead of methane.

We cited these papers in the revised manuscript (page 2, line 34 – 35, ref 9-11). However, none of these studies involved an aqueous fluid. We emphasize that experiments including water have not previously produced abundant fluid hydrocarbons heavier than methane.

5. Minor comments

Lines 1618 lithospheric diamonds (depths of at least 100 km and temperatures of 1100 C) with fluid inclusions of abiogenic methane have recently been documented (Smit et al., 2016 Lithos).

Line 27 should 'decarboxylation of isobutane' read 'decarboxylation of acetate' instead?

Smit et al, 2016 is cited in the revised manuscript (page 2, line 34, ref 7).

The revised manuscript has been changed to “decarboxylation of acetate”.

Reviewer #2 (Remarks to the Author):

Vinegar into oil: High pressure changes aqueous acetate to immiscible hydrocarbon fluid. By Huang et al. Overall, a very interesting paper with some excellent experimental data. The authors drift a little in their message and need to define an objective, choose a narrative and then maintain it from start to finish. Title: The title prepares the reader for a chemical engineering article but the introduction sets a very different scene. I would remove the “Vinegar into oil” part and add a second title that was something like “:persistent organic fluids in the subsurface”

We agree with the reviewer and have modified the title to “Immiscible hydrocarbon fluids in the deep carbon cycle”

Summary Paragraph Page 2: I think “decarboxylation of isobutene” should be “decarboxylation of acetic acid” or “decarboxylation of acetate”.

Changed to decarboxylation of acetate.

Main text Page 2: “subducted solid organic matter ...” is slightly incorrect because solid organic matter subjected to increasingly high pressures and temperatures converts to liquid and then gas. This is the concept behind production ratios. Why not say just “subducted organic matter ...”

The manuscript has been revised accordingly (page 2, line 45)

Main text Page 3: the observation of immiscible hydrocarbons must indicate that the saturation of hydrocarbons in the water is exceeded. This point is made on page 5, but then apparently forgotten on page 6 when considering mantle models. At 300 °C and high pressure water acts as an organic solvent and can dissolve hydrocarbons (e.g. Luong D., Sephton M.A. and Watson, J.S. 2015. Subcritical water extraction of organic matter from sedimentary rocks, *Analytica Chimica Acta*, 879, 48-57. doi: 10.1016/j.aca.2015.04.027.)

The reference has been cited (page 6, line 117, ref 22).

Main text Page 7: "... pressure opposes decarboxylation ..." would be supported by the hint of a mechanisms. Presumably hydrogen bonding is involved as implied for recently published data on polymers (e.g. Montgomery W, Potiszil C., Watson J.S. and Sephton M.A. Sporopollenin, a natural copolymer, is robust under high hydrostatic pressure. *Macromolecular Chemistry and Physics*, in press. doi: 10.1002/macp.201600142)

In the revised manuscript we added this reference (page 8, line 169, ref 27)

Reviewers' Comments:

Reviewer #1 (Remarks to the Author)

This revised manuscript has substantial improvements and I feel that the authors have adequately addressed reviewer queries.

The figure showing subduction PT conditions and the added discussion in the text greatly benefits the paper and I like how they have shown their own experimental conditions compared to other experimental and modelling results - this makes it easy for the reader to visualise.

I'm also satisfied that the authors addressed the issue of applicability to natural geological conditions with additional modelling, and provided examples for the reader of acetate occurrence in natural samples.

I have no further objections and can recommend that this paper be published.

Reviewer #2 (Remarks to the Author)

The paper has some interesting data that looks to have been acquired with great care and attention. It does feel as though the paper is good data looking for an idea. I am sure that the exciting story exists and that the final paper should appear in a high ranking journal.

The production of hydrocarbons from functionalized molecules is not really a dramatic observation in the context of the global carbon cycle. This process occurs naturally during conventional catagenesis. Some work is needed to communicate how your data says something really unique.

The title is much better. You still mention decarboxylation of isobutane in the abstract, when of course isobutane has no carboxyl group.

I think the paper is getting closer to where it needs to be, but needs a strong contextual idea that is then propagated from start to finish. The paper does not really flow at present.

One option is to consider the relatively rapid flux of reduced organic compounds to the deep carbon cycle. Water and pressure are known to retard organic metamorphism. Perhaps you have just demonstrated that the protective nature of these two agents leads to planetary scale consequences.

Please persevere. I think it will be an excellent paper in the end.

Reviewer #1 (Remarks to the Author):

This revised manuscript has substantial improvements and I feel that the authors have adequately addressed reviewer queries.

The figure showing subduction PT conditions and the added discussion in the text greatly benefits the paper and I like how they have shown their own experimental conditions compared to other experimental and modelling results - this makes it easy for the reader to visualize. I'm also satisfied that the authors addressed the issue of applicability to natural geological conditions with additional modelling, and provided examples for the reader of acetate occurrence in natural samples.

I have no further objections and can recommend that this paper be published.

Thank you very much.

Reviewer #2 (Remarks to the Author):

The paper has some interesting data that looks to have been acquired with great care and attention. It does feel as though the paper is good data looking for an idea. I am sure that the exciting story exists and that the final paper should appear in a high-ranking journal.

We have made a number of changes to the text in order to emphasize a specific story line: a role for immiscible hydrocarbon fluids in Earth's deep carbon cycle. These changes can be seen in the revised abstract (lines 13 – 16, and 21 – 24), as well as in the manuscript on lines 30 - 32, 39 - 47, 63 - 64, 138 - 140, 143 – 144, 182 – 188.

The production of hydrocarbons from functionalized molecules is not really a dramatic observation in the context of the global carbon cycle. This process occurs naturally during conventional catagenesis. Some work is needed to communicate how your data says something really unique.

We agree about conventional catagenesis. To address this comment we have stressed the importance of high pressure in our results. Also that the experimental product (isobutane) has a higher carbon number than that in previous experimental studies (revisions cited above).

The title is much better. You still mention decarboxylation of isobutane in the abstract, when of course isobutane has no carboxyl group.

It is fixed now (line 21 - 22).

I think the paper is getting closer to where it needs to be, but needs a strong contextual idea that is then propagated from start to finish. The paper does not really flow at present. One option is to consider the relatively rapid flux of reduced organic compounds to the deep carbon cycle. Water and pressure are known to retard organic metamorphism. Perhaps you have just demonstrated that the protective nature of these two agents leads to planetary scale consequences.

Or strong contextual idea is now as described above: a role for immiscible hydrocarbon fluids in Earth's deep carbon cycle, as emphasized in the revised text at the line numbers given above.